# Esophageal Findings in the Setting of a Novel Preventive Strategy to Avoid Thermal Lesions during Hybrid Thoracoscopic Radiofrequency Ablation for Atrial Fibrillation

**DOI:** 10.3390/jcm10214981

**Published:** 2021-10-27

**Authors:** Rani Kronenberger, Ines Van Loo, Carlo de Asmundis, Maridi Aerts, Sandro Gelsomino, Vincent Umbrain, Gian-Battista Chierchia, Mark La Meir

**Affiliations:** 1Cardiac Surgery Department, Universitair Ziekenhuis Brussel, 1090 Brussels, Belgium; rani.kronenberger@uzbrussel.be (R.K.); Ines.VanLoo@uzbrussel.be (I.V.L.); sandro.gelsomino@maastrichtuniversity.nl (S.G.); 2Heart Rhythm Management Center, Postgraduate Program in Cardiac Electrophysiology and Pacing, Universitair Ziekenhuis Brussel, 1090 Brussels, Belgium; Carlo.DeAsmundis@uzbrussel.be (C.d.A.); JeanBaptiste.Chierchia@uzbrussel.be (G.-B.C.); 3Gastroenterology Department, Universitair Ziekenhuis Brussel, 1090 Brussels, Belgium; Maridi.Aerts@uzbrussel.be; 4Anesthesiology Department, Vrije Universiteit Brussel, Universitair Ziekenhuis Brussel, 1090 Brussels, Belgium; Vincent.Umbrain@uzbrussel.be

**Keywords:** atrial fibrillation ablation, atrio-esophageal fistula, hybrid ablation, esophageal thermal lesions, esophagogastroduodenoscopy, radiofrequency

## Abstract

Purpose The development of an atrio-esophageal fistula, a rare yet potentially lethal complication of ablation for atrial fibrillation, could be related to direct tissue heat transfer during and immediately after the ablation. We therefore studied the postoperative esophageal findings by esophagogastroduodenoscopy in patients that underwent a hybrid ablation procedure using a novel preventive strategy to avoid thermal lesions. Methods Thirty-four patients (28 males; 65 years ± 9 years) were retrospectively included. All underwent a hybrid ablation in our center between April 2015 and November 2019 and agreed to an esophagogastroduodenoscopy within 0–14 days (mean: 5 days) following the ablation. To reduce the incidence of thermal lesions three procedural preventive strategies were introduced: (i) videoscopic intrathoracic transesophageal echocardiographic probe visualization to understand the relationship between posterior left atrial wall and esophagus, with probe retraction before ablation; (ii) lifting the cardiac tissue away from the esophagus during energy application; and (iii) a 30-s cool-off period after energy delivery with irrigation of the device, the ablated tissue, and the surrounding tissues. Results No esophageal thermal lesions were observed. One third of patients were diagnosed with incidental esophageal findings unrelated to the ablation procedure (11; 32.4%). Conclusion Novel preventive strategies by visualization and by avoiding contact between the ablation catheter or ablated tissue and the pericardium, seems to eliminate the potential risk of esophageal thermal lesions in the setting of hybrid ablation. Since one third of patients had preexisting esophageal disease, a more comprehensive pre-operative screening could be important to reduce the risk.

## 1. Introduction

Hybrid ablation, defined by the 2012 and 2017 HRS/EHRA/ECAS expert consensus [1,2] as a joint thoracoscopic and transvenous ablation procedure through a partnership between the surgeon and the electrophysiologist in a one- or two-stage procedure, has become a well-accepted procedure for the treatment of symptomatic atrial fibrillation (AF) in patients with persistent or long-standing persistent AF. Although this treatment modality has an excellent safety profile and low complication risk in experienced centers, the risk of the development of an atrio-esophageal fistula (AEF) could be increased when compared to endocardial and epicardial ablation alone since ablations are performed from both sides. This uncommonly encountered yet catastrophic complication of AF intervention [3] has a reported mortality rate of over 70% [4]. The incidence is estimated at 0.03–0.08% for both endocardial and epicardial ablation. However, this might be underestimated due to underreported cases and misdiagnosis [2,5,6,7]. A potential advantage of an epicardial ablation technique is that the delivered energy is directed from the outside of the heart towards the endocardium, thus potentially avoiding heat transfer to the esophagus or other adjacent cardiac structures. However, during epicardial application of radiofrequency (RF) energy on the inferior line, from the right inferior pulmonary vein (PV) towards the left inferior PV, heat transfer to the esophagus could be possible if there is direct contact between the ablation device or the ablated tissue with the posterior pericardium overlying the esophagus. This can still occur if the structures are not sufficiently separated from each other due to insufficient lifting of the ablation tool towards the heart, hence, away from the pericardium. In contrast, when performing an epicardial roof line ablation by connecting the superior PVs just below the Bachmann bundle, typically in a more anterior position compared to an endocardial roof line, little risk of thermal injury is to be expected since there is a sufficient space between the ablation tool and the esophagus. This spatial buffer is in fact the lower (posterior) part of the roof of the left atrium. As a result, the left atrial roof line performed from the epicardium is never in close contact with the esophagus, and as such, not a risk for AEF.

In the present study, we aimed to describe the esophageal findings in a series of patients presenting with symptomatic atrial fibrillation that underwent a hybrid procedure. Furthermore, we evaluated the impact of three safety measures to prevent potential damage of the esophagus.

## 2. Materials and Methods

### 2.1. Study Population

Thirty-four patients treated for symptomatic AF from April 2015 to November 2019 agreed to a postoperative esophagogastroduodenoscopy (EGD) to assess for esophageal thermal lesions (ETL) after having undergone an epicardial thoracoscopic ablation with, if needed, a periprocedural endocardial touch-up.

### 2.2. Pre-Procedural Management

All patients provided written informed consent for their evaluation and research participation. The design of this study was approved by the ethical committee. Prior to the procedure, left atrial and pulmonary vein (PV) imaging was performed, including transthoracic echocardiogram (TTE) and cardiac computed tomography (CT), as well as an electrocardiography (ECG) and pulmonary function tests. In addition, a perioperative transesophageal echocardiography (TEE) was performed in order to exclude a possible thrombus in the left atrium.

### 2.3. The Epicardial Ablation Procedure

The procedure was performed in the hybrid operating room as previously described [8]. In brief, the procedure was carried out under general anesthesia and patients were intubated by a double-lumen EZ-Blocker (Teleflex Inc., Morrisville, NC, USA) for selective lung ventilation. The epicardial ablation comprised bilateral antral PV isolation with 4 to 6 applications using a bipolar, bidirectional RF clamp (Isolator Synergy Ablation Clamp; AtriCure, Inc, Mason, OH, USA). In addition, all patients received an identical epicardial posterior “box” lesion set, comprising a roofline (connecting the superior PVs) and floor line (connecting the inferior PVs) using the bipolar unidirectional RF rail (Coolrail; AtriCure, Inc., Mason, OH, USA) or linear pen device (Isolater Pen and Coolrail, Atricure, Mason, OH, USA) to isolate the posterior LA. The ganglionated plexi were ablated, as well as the ligament of Marshall. Entrance and exit blocks were confirmed if the patient was in sinus rhythm. In all patients, the left atrial appendage was excluded using a clipping device. (AtriClip, AtriCure, USA, OH, USA). The EP procedure was performed by electroanatomic mapping of the LA with a non-fluoroscopic navigation system (CARTO system, Biosense Webster, Inc. Diamond Bar, CA, USA or Ensite Precision, Abbott, St Paul, MN, USA) using the circular mapping catheter (Lasso, Biosense Webster, Inc., Irvine, CA, USA). If incomplete lesions were found, an open-irrigated 3.5-mm tip RF ablation catheter (NaviStar ThermoCool; Biosense Webster, Inc., Irvine, CA, USA) was used for touch-up. A cavotricuspid isthmus (CTI) was created when an atrial flutter was present or occurred during the ablation. If a mitral isthmus-dependent flutter developed during the procedure, a mitral line was created. In addition, patients who continued to remain in AF, left and right complex fractionated atrial electrograms (CFAEs) were mapped and ablated.

### 2.4. Preventive Strategies to Avoid Esophageal Damage

Three procedural safety measures were conducted with the aim of preventing potential damage to the esophagus, including (i) videoscopic intrathoracic transesophageal echocardiographic probe visualization to understand the relationship between posterior left wall and esophagus and then withdrawing the TEE probe to a more cephalad position in the esophagus to twenty centimeters away from the incisors before RF energy delivery; (ii) lifting the ablation tool and the atrial tissues away from adjacent structures during energy delivery; and (iii) implementing a 30 s cool-off period with lifting of the ablated tissue, and simultaneous irrigation of the ablation device, the ablated tissue, and adjacent structures after ending the energy application.

### 2.5. Postprocedural Management

After the ablation procedure, patients were monitored at the intensive care unit (ICU). Before discharge, they underwent a transthoracic echocardiogram in order to exclude a post-operative pericardial effusion. Low-molecular-weight heparin was started 6 h after the ablation, and oral anticoagulation or non-vitamin K antagonist Oral Anti-Coagulants was reinitiated on the 4th postoperative day. Oral anticoagulation and antiarrhythmic drugs were continued for at least 3 months in all patients. A proton pump inhibitor (PPI) was added for 2–4 weeks.

### 2.6. Esophageal Evaluation

Esophageal and gastric endoscopic examination was done within 0–14 days post-ablation (mean: 5 days). ETL were defined as lesions varying from erythema to ulcerations that could have been provoked by the ablation. All lesions were carefully assessed by an experienced gastroenterologist. Subsequent microscopic work-up was performed in ten patients (29.4%).

### 2.7. Follow-Up and Surveillance

Three weeks after discharge, a follow-up cardiac consultation was planned. Additionally, in patients with esophageal lesions in which follow-up was deemed necessary, an additional EGD was planned.

Before discharge, patients were educated extensively on the signs and symptoms of esophageal injury (Figure 1) to ensure postoperative clinical vigilance. The protocol for patients with a high index of clinical suspicion for AEF in the weeks after the ablation procedure consisted of a full emergent diagnostic work-up comprising a blood analysis, TTE, and contrast-enhanced CT of the chest, endoscopic manipulation should be highly discouraged in these patients.

### 2.8. Statistical Analysis

Data were retrospectively entered into a database. Normally distributed continuous variables are expressed as mean ± standard deviation, unless otherwise specified. Categorical variables are expressed as numbers and percentages.

## 3. Results

### 3.1. Baseline Population Characteristics

Thirty-four patients (28 males, 65 years ± 9 years) were retrospectively included in the study. Mean BMI was 27.6 (range 20.4–34.7). The indication for the ablation procedure was paroxysmal AF in nine patients (26.5%), persistent AF in 17 patients (50%) and long-standing persistent AF in the remaining eight (23.5%). All patients had failed ≥1 Class I or III antiarrhythmic drugs. Mean left atrial size was 4.47 ± 0.76 cm. None of the patients had significant coronary artery disease, COPD > GOLD II, or diastolic heart failure. The left ventricular ejection fraction (LVEF) was >50% in the majority of patients (94.1%). Comorbidities included previous transient ischemic attack (TIA) or stroke (7; 20.6%), hypertension (21; 61.8%), diabetes (3; 8.8%), and sleep apnea (3; 8.8%). Fifteen patients (44.1%) were previously cardioverted and 22 patients (64.7%) had ≥1 previous endocardial catheter ablation (CA). Gastroesophageal reflux disease (GERD) was reported in three patients (8.8%). Baseline characteristics are listed in Table 1.

### 3.2. Procedural Characteristics

All patients (*n* = 34) had a hybrid ablation procedure. Twenty-five patients (73.5%) underwent a unilateral left-sided thoracoscopy, whereas nine patients (26.5%) underwent a bilateral thoracoscopy. In 15 of these 34 patients (44.1%), endocardial touch-ups were required. One patient (2.9%) needed roof line completion to complete the posterior box. Complex fractionated atrial electrogram (CFAE) mapping and ablation was performed in eight patients (23.5%). A mitral isthmus, as well as a cavotricuspid isthmus (CTI) ablation was performed in seven (20.6%) and seven (20.6%) patients, respectively. Seventeen patients (50%) were administered PPI before the ablation, and all patients received PPI for 2–4 weeks post-ablation. Procedural characteristics are listed in Table 2.

### 3.3. Procedural Complications

There were no peri-procedural complications. During follow-up, no major complications, including death, cerebrovascular events, AEF, and pacemaker insertion occurred.

### 3.4. Esophageal Findings

Esophageal thermal lesions were found in none of the patients. Other incidental esophageal findings were found in 11 patients (32.4%), sometimes with multiple findings in one patient, including reflux esophagitis (3; 8.8%), metaplasia (3; 8.8%), ulcers (2; 5.9%) of which one was iatrogenic due to TEE manipulation, and one was related to GERD. Furthermore, Barrett’s esophagus (2; 5.9%) and erosions (1; 2.9%) were found, as well as TEE-induced ecchymosis (1; 2.9%). Further on, glycogen acanthosis (1; 2.9%) and Candida albicans infection (1; 2.9%) were observed. Detailed esophageal findings are listed in Table 3. Gastric lesions include antritis (5; 14.7%), gastritis (5; 14.7%), erosions (3; 8.8%), hiatal hernia (3; 8.8%), fundic gland polyps (3; 8.8%), ulcerations (2; 5.9%), cardia mucosal edema (1; 2.9%), Cameron lesions (1; 2.9%), and metaplasia (1; 2.9%). Anomalies in the duodenum included inflammation (4; 11.8%), ulcerations (2; 5.9%), erosions (1; 2.9%), diverticle (1; 2.9%), and a tubular adenoma with low grade dysplasia (1; 2.9%). Of the 17 patients that took PPI preoperatively, six patients (35.3%) had esophageal anomalies. Similarly, of the 17 patients that did not receive PPI, five patients (29.4%) had esophageal anomalies.

### 3.5. Follow-Up

At a mean follow up of 47.5 ± 19.2 months (median 55.7), no esophageal thermal lesions or fistula’s were reported. Of the patients that were diagnosed with incidental esophageal findings (11; 32.4%), three follow-up EGDs (27.3%) were performed for hiatal hernia, metaplasia, and TEE-induced ulcerations.

## 4. Discussion

Whereas the exact mechanism of AEF remains unknown, esophageal thermal lesions as a direct result of ablation within the left atrium (LA) have been proposed to be the starting point [4]. The thermal insult initiates an inflammatory cascade leading to tissue necrosis. When radiofrequency heats the posterior left atrial wall, conductive heating of the anterior esophagus occurs leading to thermal injury to the collagen, elastin, and proteins thereby decreasing tensile strength of esophageal tissue. Injury to intraluminal arteries, resulting in focal ischemia and necrosis can form esophageal ulcerations, esophageal perforation, and AEF. It seems that heat damage results from thermal conduction from the tissue rather than direct power application. Since most cases of atrio-esophageal fistulae are observed between the second and fourth week (range 2–60 days), there seems to be progressive tissue necrosis with lesion expansion that may contribute to fistula formation. In addition, damage to the peri-esophageal vagal nerves can be associated with impairment of LES tonus and gastroparesis, thereby leading to gastroesophageal reflux, and pathogenesis of esophageal ulceration, which could be at the base of fistula formation. Nonetheless, no specific patient or technique related risk factors have been described for early versus late fistula formation in the literature.

Invasive treatment of atrial fibrillation using radiofrequency ablation tools requires direct contact of the catheter with the left atrium and the pulmonary veins. The close anatomical relationship between these structures and the esophagus plays a pivotal role in the risk for esophageal thermal lesion formation. Moreover, the esophagus is the only gastrointestinal organ that lacks an outer serosal layer; therefore, its vulnerability to thermal insults is increased. Understanding the anatomical relationship between the tissues to be ablated and the esophagus is crucial. The thoracic portion of the esophagus follows a route through the supero-posterior mediastinum and extends to the esophageal hiatus through the diaphragm, posterior to the LA [9]. The esophagus and the LA share a mean contact length of 42 ± 7 mm (range 30–53 mm). Sanchez-Quintana et al. described that in 40% of cases the atrio-esophagus interface was <5 mm [10]. However, the fibrofatty tissue between the esophagus and the LA is nonuniform and varies individually with age, sex, BMI, and duration of AF. The mean atrio-esophageal distance along the superior level is 2.3 ± 1.2 mm (range, 1 to 8.2 mm) from an endocardial standpoint.

Twenty-five patients (73.5%) underwent a unilateral left-sided thoracoscopy, whereas nine patients (26.5%) underwent a bilateral thoracoscopy. In 15 (44.1%), endocardial touch-ups were required. Therefore, two ablation strategies (with or without endocardial touch-ups) were performed in this patient population. In percutaneous endocardial ablation, unipolar RF energy is directed towards the posterior LA wall and the esophagus. In contrast, in epicardial ablation, bipolar RF energy is directed away from these structures. Therefore, the risk of thermal injury is expected to be lower in epicardial ablation. However, the incidence of AEF described in literature for both techniques is similar, suggesting that the emergence of an AEF is a more complex process and may not depend on the direction of the RF energy alone. Regarding complications in general, when comparing thoracoscopic surgical ablation to catheter ablation for atrial fibrillation major complications were significantly higher in the surgical ablation group (28.2 vs. 7.8%), mainly driven by pleural effusion and pneumothorax. Additionally, in hybrid versus catheter ablation, hybrid ablation had a slightly higher complication rate than catheter ablation.

Data on the rare but severe complication of atrio-esophageal fistula following epicardial RF ablation are limited. Few clinical publications are available in PUBMED and searching medical device reports via the Manufacturer and User Facility Device Experience (MAUDE) database did not provide any data on AEF with this surgical approach. Nevertheless, the MAUDE database reported 11 cases of AEF between 2015 and 2021 in thoracoscopic AF ablation using the same technology and surgical technique as described here. Events reported in MAUDE are not independently verified and does not capture the total number of procedures that are performed with a technology, it is therefore not possible to calculate the incidence of a complication. Studies regarding the occurrence of ETL after RF ablation reported esophageal wall changes in 47% of patients after undergoing CA with 29% of patients assessed with erythema and 18% with ulcer-like changes [11]. Similarly, Singh et al. reported nine of 81 patients (11%) with esophageal ulcerations after a CA procedure [12]. Since no ETL were seen in this study, we assume that three particular preventive measures are efficient in preventing thermal injury in a hybrid AF procedure: (i) transesophageal echocardiographic probe visualization with probe retraction before ablation, (ii) lifting the cardiac tissue during energy application, and (iii) a 30-s cool-off period after energy delivery with lifting of the ablated tissue and simultaneous irrigation of the device, the ablated tissue, and the surrounding tissues.

The possible electrical and thermal interactions between a TEE probe and an RF ablation catheter have not been extensively studied. As the TEE probe is positioned directly behind the LA, the electrical charge over the head of the probe could interact with the ablation catheter (especially if it is unipolar) and produce excessive heat in the area. In one study, computer simulations using a theoretical model in order to investigate the effect of different factors on the temperature distributions in the esophagus during ablation were performed. The authors concluded that esophageal injury is exclusively due to thermal conduction from the atrium, and mainly influenced by the thickness of connective tissue [13]. Additionally, the possibility that thermal latency can occur in the esophagus where temperature could continue to rise even after RF delivery has been discontinued, has not been well studied. An important thermal latency between atrial RF delivery and atrial tissue temperature has been described [14]. Heat transfer by conduction is a slow physical phenomenon that requires 10–30 s to enlarge over time. Although the thermal lesion boundaries grow slowly over time, temperature can augment rapidly in sites 1–2 mm away from the zone where electrical power is converted into heat. In this model, high esophageal temperatures were observed when the distance between the ETP and the ablation catheter were <3 mm. In this scenario, temperature will increase rapidly due to thermal conduction without any electrical interaction. Interestingly, in our analysis, the ETL complication rate (0%) was lower than both Schmidt et al. and Singh et al. We believe that this difference in ETL is related to the fact that the ablation was for most part directed from the epicardium towards the endocardium using a bipolar catheter. Also, the preventive strategies undertaken during the ablation might prove to be of value in avoiding ETL for every thoracoscopic ablation procedure.

Several studies recommend taking patient vulnerability to thermal injury into account. The limited published data on AEF after surgery does not allow for any conclusions on risk factor characteristics of the patients and the performed surgery (ablation tools, lesion set). The likelihood of ETL could be increased in patients for whom LA is enlarged (>60 mm in diameter), who have persistent AF, and a low or normal BMI. These conditions reflect the amount of fibrofatty tissue and thus the capacity of heat conduction. Rillig et al. found that patients undergoing CA with a BMI under 26 were at higher risk for ETL, further supporting the above hypothesis [15]. The mean BMI in our patient group was 27.6 (range 20.37–34.72). Similarly, older, smaller and frail patients with thinner pericardium are more prone to injury. Recently, real-time luminal esophageal temperature (LET) monitoring has been proposed for ETL risk reduction. Halm et al. described 14.6% ulcer-like or hemorrhagic thermal lesions in patients with a risen LET during CA and correlated a 1 °C rise in LET with an increase in odds of ETL and a factor of 1.36 [16]. However, LET monitoring is not without controversy. Limitations include the sensitivity of the sensor and the challenge to achieve adequate contact between the sensor and esophageal wall. Suboptimal orientation of the temperature sensor could create an offset between true intramural tissue temperatures and intraluminal temperature as measured by the sensor. Moreover, a latency time may exist, due to a slow rise in temperature. Finally, Singh et al. reported that the use of LET temperature probes may even be detrimental by serving as a thermal conductor [12]. Other suggested preventive measures for AEF at the level of the esophagus include cooling or posterior displacement of the esophagus with the TEE probe or deviation device, and prophylactic proton pump inhibitor use. No single esophageal protective measure has proven to be sufficiently effective to eliminate the risk; therefore, esophageal protection strategies remain uncertain.

In a retrospective study of 425 patients, Knopp et al. investigated incidental and ablation-induced findings during upper gastrointestinal endoscopy in patients after percutaneous ablation of AF. Injuries seen on endoscopy were erosions (22%), ulcerations (4–60%), and thermal injuries (11%) [17]. Injuries seen on endoscopy related to the TEE probe were studied by Kumar et al. Seventy-six patients undergoing PVI with TEE, PVI/TEE, 16 undergoing PVI without TEE (PVI/No TEE), and 27 undergoing TEE without any left atrial ablation (TEE/No LA ablation) under GA were included. Esophageal lesions were seen in 30% of patients undergoing PVI with TEE, in 22% undergoing TEE without left atrial ablation (22%), and none of the patients who had PVI without TEE (0%) [18]. Therefore, excessive transesophageal echocardiographic probe manipulation should be avoided.

Patients with known severe gastroesophageal conditions should be treated more cautiously or be excluded for left atrial ablations. In accordance with the 2017 guidelines on Catheter and Surgical Ablation of Atrial Fibrillation, it is justified to prescribe short-term PPIs or histamine-2 receptor antagonists (H_2_RAs) as a routine prophylactic treatment in all patients undergoing AF ablation, until 1 to 4 weeks post-ablation. PPI ensures a pH level above 4, thus allowing healing of esophageal injury and avoiding lesion enlargement [19]. Here, no conclusions about the clinical value of PPI in reducing thermal injury could be made given the absence of ETL. Since gastroesophageal comorbidities may add to the likelihood of ETL, and the high occurrence of collateral esophageal findings in this study, a more comprehensive pre-operative screening by EGD of preexisting esophageal conditions could be important.

Clinical vigilance in the postoperative period remains the most crucial asset in preventing devastating outcomes, particularly in patients with a high clinical index of suspicion for AEF. It is imperative that before discharge, patients are educated extensively upon the signs and symptoms of ETL. Low clinical awareness and subsequently a late diagnosis indeed are predictors of mortality [4].

### Limitations

This is a single-center observational and retrospective study with a small sample size. It therefore does not allow for safety conclusions regarding the risk for esophageal thermal lesions related to hybrid ablation. Prospective monitoring of this rare adverse event in clinical trials in a large number of patients is required to accurately measure the incidence and to study the effect of the mitigation strategies. Furthermore, no literature data is available on postoperative EGD after thoracoscopic or hybrid ablation without the proposed mitigation strategies. Importantly, we did not use EGDs as a routine prophylactic endoscopic examination although this data is necessary to understand lesion progression or formation. A perioperative EGD prior to TEE and a perioperative EGD post ablation, after removing the TEE, would have added useful information on the impact of TEE manipulation on lesion formation. Intraluminal temperature in the esophagus was not recorded; this could have helped us to better understand the potential for thermal insults. Since a larger study with pre- and postoperative EGD is required to better understand and estimate the risk of esophageal injury, with or without pre-existing lesions, in AF ablation, the findings of this study should be taken as preliminary.

## 5. Conclusions

In contrast to other papers describing the risk of esophageal thermal lesions during AF ablation, no thermal esophageal lesions were found in this hybrid study. Therefore, preventive strategies should be performed to eliminate the potential risk of esophageal thermal lesions in the setting of hybrid ablation by (i) visualization of the anatomical relationship between the esophagus and the posterior wall of the left atrium, (ii) avoiding contact between the ablation catheter or ablated tissue and the pericardium, and (iii) implementing a 30-s cool-off period after energy delivery with lifting of the ablated tissue and simultaneous irrigation of the device, the ablated tissue, and the surrounding tissues. Since a significant number of patients programmed for a hybrid AF ablation procedure have preexisting esophageal disease, which may add to the likelihood of thermal and mechanical insults, a more comprehensive pre-operative screening could be warranted in this patient population.

## Figures and Tables

**Figure 1 jcm-10-04981-f001:**
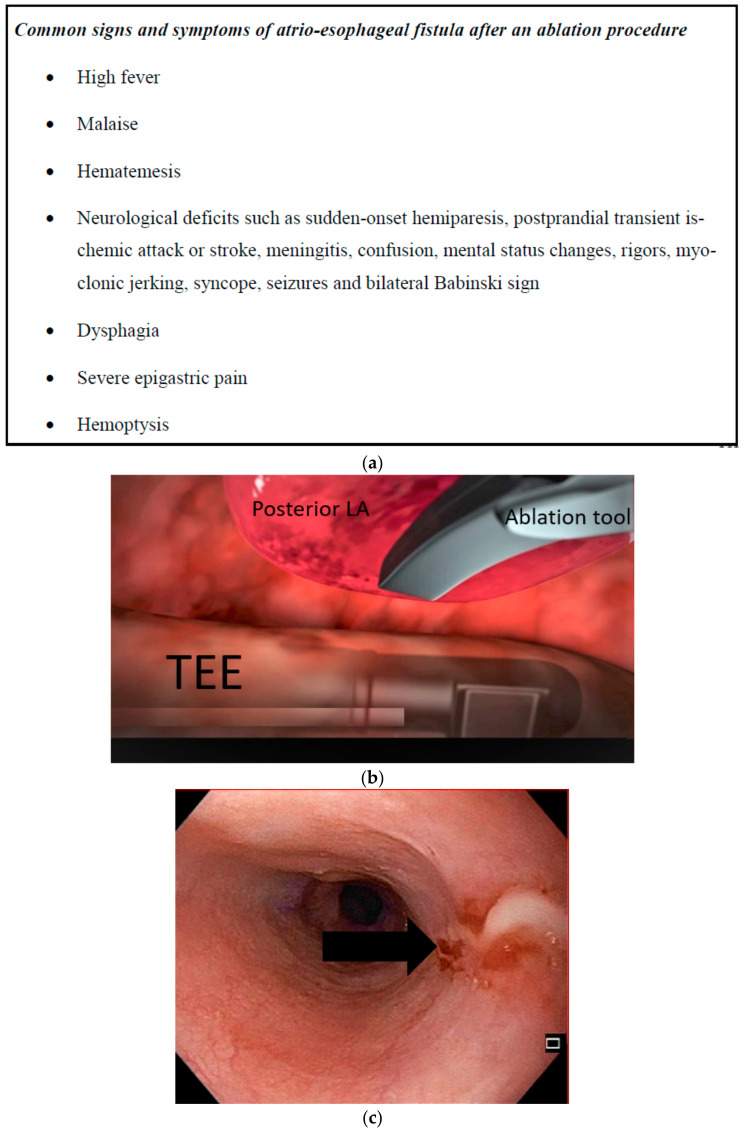
(**a**) Patient esophageal symptoms awareness brochure. (**b**) Anatomical relation between esophagus with TEE probe and posterior left atrium lifted by an epicardial ablation tool. (**c**) Laceration in the esophagus by the TEE probe.

**Table 1 jcm-10-04981-t001:** Baseline characteristics.

Baseline Characteristics		*n =* 34
Male gender		28 (82.4)
Age at enrollment		65 years ± 9 (47–82)
BMI		27.6 (20.37–34.72)
Paroxysmal AF		9 (26.5)
Persistent AF		17 (50)
Long standing persistent AF		8 (23.5)
Left atrial size (cm)		4.47 ± 0.76
LVEF (%)		55.9 ± 4.8
LVEF	>50%	32 (94.1)
	30–50%	2 (5.9)
	<30%	0
Diabetes		3 (8.8)
Hypertension		21 (61.8)
Sleep apnea		3 (8.8)
COPD		0 (0)
Coronary artery disease		0 (0)
Previous TIA/stroke		7 (20.6)
Systolic heart failure		0 (0)
Previous cardioversion		15 (44.1)
Previous CA		22 (64.7)
GERD		3 (8.8)

Categorical variables are expressed as absolute and percentage (in brackets). Continuous variables are expressed as mean + SD. Values are *n* (%); Abbreviations: AF = atrial fibrillation; BMI = body mass index; CA = catheter ablation; COPD = chronic obstructive pulmonary disease; GERD = gastroesophageal reflux disease; TIA = transient ischemic attack; LVEF = left ventricular ejection fraction.

**Table 2 jcm-10-04981-t002:** Procedural characteristics.

Procedural Characteristics	*n =* 34
Epicardial ablation	34 (100)
Endocardial ablation	24 (70.6)
CFAEs	8 (23.5)
Roofline	1 (2.9)
Mitral line	7 (20.5)
CTI-line	7 (20.5)
Unilateral VATS	25 (73.5)
Bilateral VATS	9 (26.5)

Values are *n* (%); Abbreviations: CFAE complex fractionated atrial electrograms; CTI cavotricuspid isthmus; VATS video assisted thoracoscopic surgery.

**Table 3 jcm-10-04981-t003:** Esophageal findings on EGD and biopsy.

Esophageal Findings on EGD and Biopsy	
Reflux-esophagitis	3 (8.8)
Metaplasia	3 (8.8)
Ulceration	2 (5.9)
GERD-induced	1 (2.9)
TEE-induced	1 (2.9)
Barrett’s esophagus	2 (5.9)
Erosion	1 (2.9)
TEE-induced ecchymosis	1 (2.9)
Glycogen acanthosis	1 (2.9)
Candidiasis	1 (2.9)

Values are *n* (%). Abbreviations: GERD gastroesophageal reflux disease; TEE transesophageal echocardiography.

## Data Availability

The data presented in this study are available on request from the corresponding author. The data are not publicly available due to GDPR privacy restrictions.

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
