# Peer review of "Esophageal Findings in the Setting of a Novel Preventive Strategy to Avoid Thermal Lesions during Hybrid Thoracoscopic Radiofrequency Ablation for Atrial Fibrillation"

_jcm, 2021, doi:10.3390/jcm10214981_

Round 1
Reviewer 1 Report
Thank you very much for submitting the manuscript: "Esophageal findings in the setting of a novel preventive strategy to avoid thermal lesions during hybrid thoracoscopic radiofrequency ablation for atrial fibrillation."
The atrio - esophageal fistula is a complication specific to ablation procedures related to the treatment of atrial fibrillation. This complication was first described in 2004, and the number of similar reports has increased since then. Literature data show that the formation of a fistula may occur even 38 days after the ablation procedure. Are there data on risk factors for fistula formation "early" after ablation ie 3 - 5 days and later, ie after 20 - 30 days ? What strategy could be proposed for "late" fistulas ?
Author Response
Dear reviewer,
We first would like to thank you for considering our paper for publication and for allowing the constructive discussion in this round of review.
Your comments provided valuable insights to improve the content of our paper. We tried to address your remarks as best as possible. Each of the comments are numerically listed. As requested, no new text was uploaded since a new version of the manuscript will only be available once all the reviewer comments have been replied to.
Question 1: The atrio - esophageal fistula is a complication specific to ablation procedures related to the treatment of atrial fibrillation. This complication was first described in 2004, and the number of similar reports has increased since then. Literature data show that the formation of a fistula may occur even 38 days after the ablation procedure. Are there data on risk factors for fistula formation "early" after ablation ie 3 - 5 days and later, ie after 20 - 30 days? What strategy could be proposed for "late" fistulas?
Anwer 1: Atrio-esophageal fistula formation after LA ablation with RF was initially described during open heart surgery. Later, similar fatal complications were observed after catheter-based ablation for the treatment of atrial fibrillation. Since most cases of atrio-esophageal fistulae are observed between the second and fourth week (range 2-60 days), there seems to be progressive tissue necrosis with lesion expansion that may contribute to fistula formation. One wonders whether the lesion should be present on both sides of the esophageal muscularis layer or whether the lesion progresses from the luminal endothelial side towards the atrium. In an acute animal study by Aupperle et al, Ablation of atrial fibrillation and esophageal injury: Effects of energy source and ablation technique, J Thorac Cardiovasc Surg 2005;130:1549-54, doi:10.1016/j.jtcvs.2005.06.052, the authors created circular lesions endocardially or epicardially in the left atrium and at the pulmonary veins by using different energy sources: cryoablation, microwave, laser, and unipolar or bipolar radiofrequency in 39 sheep. Temperatures inside the esophagus were measured and esophageal tissue was investigated macroscopically and histopathologically. Esophageal damage was seen histologically in 24 of 39 cases, interestingly, the epithelial layer remained intact. The authors suggested that vascular damage may induce ischemic alterations of the epithelial layer, thereby increasing the risk of perforation.
To the best of our knowledge, there are no data on the chronic changes of the esophageal lesions histologically describing progression of the lesion over time after ablation in animal models.
We could not find specific patient-related risk factors for early fistula formation compared to late formation in the literature. We suppose that this difference is related to a direct energy transfer into the esophagus during the ablation in early formation and supposedly only indirect heat transfer in the later formation. The cumulative amount of energy locally delivered, resulting from an interplay of power, time, and contact‐force (CF), has probably been directed immediately towards the esophagus in early fistulas compared to later fistulas were the lesion may have progressed from a more indirect energy transfer. We therefore believe that the best strategy to avoid late fistulas is avoidance of indirect heat transfer. In addition, damage to the peri-esophageal vagal nerves can be associated with impairment of LES tonus and gastroparesis, thereby leading to gastroesophageal reflux, and pathogenesis of esophageal ulceration, which could be at the base of fistula formation.
We propose to add the following in the discussion:
Since most cases of atrio-esophageal fistulae are observed between the second and fourth week (range 2-60 days), there seems to be progressive tissue necrosis with lesion expansion that may contribute to fistula formation. In addition, damage to the peri-esophageal vagal nerves can be associated with impairment of LES tonus and gastroparesis, thereby leading to gastroesophageal reflux, and pathogenesis of esophageal ulceration, which could be at the base of fistula formation. Nonetheless, no specific patient-related risk factors have been described for early versus late fistula formation in the literature.
Reviewer 2 Report
Authors aimed to present an esophagal findings after the catheter ablation.They found on thermal lesions after the procedure. However no gastroscopy was performed prior the ablation. This study is also low population research; I am cowries to see the pre-study power analysis.
This study present the results of using the novel ablation strategy. I am missing the comparison with traditional ablation procedures with risk factor identification and analysis of esophageal thermal lesions.
Author Response
Dear reviewer,
We first would like to thank you for considering our paper for publication and for allowing the constructive discussion in this round of review.
Your comments provided valuable insights to improve the content of our paper. We tried to address your remarks as best as possible. Each of the comments are numerically listed. As requested, no new text was uploaded since a new version of the manuscript will only be available once all the reviewer comments have been replied to.
Question 1: Authors aimed to present an esophageal findings after the catheter ablation. They found no thermal lesions after the procedure. However no gastroscopy was performed prior the ablation. This study is also low population research; I am curious to see the pre-study power analysis.
Answer 1: Thank you for making this valuable point. The study was designed after having obtained information in 2015 from one center that experienced 3 AEFs out of 21 patients in minimally invasive video-assisted epicardial beating heart ablation for lone atrial fibrillation, later published by Kik et Al. Thorac Cardiovasc Surg 2017; 65(06): 472-472. DOI: 10.1055/s-0036-1597913. Since the three patients became symptomatic between 5 and 8 weeks post-procedure we supposed that the lesions weren’t caused by direct energy delivery from the catheter towards the esophagus but indirectly by thermal heating and progressive tissue necrosis with lesion expansion. Although the reported incidence in the literature was only three cases until this paper, we deemed it important to investigate this potentially lethal complication further. Due to the rarity of AEF (0.03-0.08%), the numbers of patients to be included in the study are deemed unreasonable for this treatment option. 11% ETL were found on EGD in the largest retrospective study of 425 patients in 2014 by Knopp et al. Therefore, we decided to study the incidence of ETL as a potential precursor for AEF in hybrid procedures.
To better understand the mechanism of thermal lesions we also measured the temperature at the side of the CoolRail ablation catheter intraoperatively. Interestingly, we found that these temperatures systematically went above 50°C, as well as the temperature of the ablated tissue (50°C was the upper range limit of the sterile T° probe). We then went to the lab for an in-vitro study where we ablated left atrial tissue inside the ABLA-BOX (The ABLA-BOX: An In Vitro Module of Hybrid Atrial Fibrillation Ablation, Lozekoot et al., Innovations (Phila). May-Jun 2016;11(3):201-9. doi: 10.1097/IMI.0000000000000256.) and measured temperatures of certain areas to nearly 100°C on both the ablation tool and the ablated tissue (unpublished data). Therefore we concluded that it is crucial to avoid any immediate direct contact between the epicardial CoolRail, the ablated tissue and the posterior pericardium before active cooling of both the device and the tissue. Once this working strategy was decided upon, it was thought unethical to perform ablations without these mitigation strategies to minimize the risk of AEF. Therefore no data is available in the literature on EGD after epicardial ablation without the mitigation strategies. We believe that there would have been a reasonable chance of finding an ETL in 34 patients if there were no mitigation strategies applied.
Question 2: This study present the results of using the novel ablation strategy. I am missing the comparison with traditional ablation procedures with risk factor identification and analysis of esophageal thermal lesions.
Answer 2: Nair et al. reported five cases of proven AEF (0.07 %) in “The prevalence and risk factors for atrioesophageal fistula after percutaneous radiofrequency catheter ablation for atrial fibrillation: the Canadian experience”, J Interv Card Electrophysiol. 2014 Mar;39(2):139-44. doi: 10.1007/s10840-013-9853-z. Risk factor identification revealed that operators who reported AEF more often used general anesthesia and were also more likely to be users of the non-brushing technique in the posterior wall of the LA. In general, use of general anesthesia, ablation time, temperature and power settings, and ablation over the posterior wall are hypothesized as potential contributors. Suggested preventive measures for AEF at the level of the esophagus include temperature monitoring, cooling or posterior displacement of with the TEE probe or deviation device, and prophylactic proton pump inhibitor use. No single esophageal protective measure has proven to be sufficiently effective to eliminate the risk, therefore esophageal protection strategies remain uncertain.
Analysis of esophageal thermal lesions: It is postulated that the process of fistulization is initiated in the esophagus from thermal ulceration and progress towards LA. The mechanisms involved in this process are not entirely understood. The use of a standardized classification of esophageal lesions, Kansas City Classification [KCC]), was recently proposed to describe the lesions and to help as a decision making tool for treatment strategies.
Regarding the limited published data on AEF after surgery, we weren't able to make any conclusions on risk factors characteristics of the patients and their surgery. Since Kik et al. reported an extremely high incidence of fistula's, their surgical technique may have been responsible for their series.
We thank you for your excellent comments and hope we have answered your questions accordingly.
Reviewer 3 Report
The authors retrospectively studied 34 patients who underwent a hybrid ablation between April 2015 and November 2019 and agreed to an esophagogastroduodenoscopy within 0-14 days (mean: 5 days) following the ablation. To reduce the incidence of thermal lesions three procedural preventive strategies were introduced, (i) videoscopic intrathoracic transesophageal echocardiographic probe visualization to understand the relationship between posterior left atrial wall and esophagus, with probe retraction before ablation (ii) lifting the cardiac tissue away from the esophagus during energy application, and (iii) a second cool-off period after energy delivery with irrigation of the device, the ablated tissue, and the surrounding tissues. The authors did not find any esophageal lesions in these patients. I have a few comments and suggestions for the authors:
- The authors provide information of incidence of AEF after endocardial and epicardial ablation. However, not much information is provided on the incidence of AEF after the hybrid approach. What is the incidence of AEF after hybrid AF ablation? Is there any prior report of occurrence of this complication after hybrid approach. If yes, this data perhaps should be included in the introduction section.
- Why was the hybrid approach instead of the endocardial approach chosen for these patients? It seems that 12 patients had not had any prior ablation procedure.
- What are the risks of this approach vs epicardial vs endocardial AF ablation. The readers should be provided some sense of the relative risks of the three approaches.
- Thoracoscopy provides direct visuals, which endocardial and epicardial CA does not provide. This factor alone might lead to a decrease in esophageal injury. It might therefore be difficult to conclude that measures that the authors instituted led to a decrease in esophageal injury. Overall the numbers are small so it may be quite difficult to draw any firm conclusions from this data.
Author Response
Dear reviewer,
We first would like to thank you for considering our paper for publication and for allowing the constructive discussion in this round of review.
Your comments provided valuable insights to improve the content of our paper. We tried to address your remarks as best as possible. Each of the comments are numerically listed. As requested, no new text was uploaded since a new version of the manuscript will only be available once all the reviewer comments have been replied to.

Round 2
Reviewer 3 Report
The authors have addressed my comments satisfactorily. I have no additional comments.